# Teleost Eye Is the Portal of IHNV Entry and Contributes to a Robust Mucosal Immune Response

**DOI:** 10.3390/ijms25010160

**Published:** 2023-12-21

**Authors:** Xinyou Wang, Guangyi Ding, Peng Yang, Gaofeng Cheng, Weiguang Kong, Zhen Xu

**Affiliations:** 1Department of Aquatic Animal Medicine, College of Fisheries, Huazhong Agricultural University, Wuhan 430070, China; wxy1998518@163.com; 2Key Laboratory of Breeding Biotechnology and Sustainable Aquaculture, Institute of Hydrobiology, Chinese Academy of Sciences, Wuhan 430072, China; 202071741@yangtzeu.edu.cn (G.D.); chenggaofeng@ihb.ac.cn (G.C.); kongweiguang@ihb.ac.cn (W.K.)

**Keywords:** mucosal immune system, ocular mucosa, viral infection, RNA-seq, *Oncorhynchus mykiss*

## Abstract

The ocular mucosa (OM) is an important and unique part of the vertebrate mucosal immune system. The OM plays an important role in maintaining visual function and defending against foreign antigens or microorganisms, while maintaining a balance between the two through complex regulatory mechanisms. However, the function of ocular mucosal defense against foreign pathogens and mucosal immune response in bony fish are still less studied. To acquire deeper understanding into the mucosal immunity of the OM in teleost fish, we established a study of the immune response of rainbow trout (*Oncorhynchus mykiss*) infected with the infectious hematopoietic necrosis virus (IHNV). Our findings revealed that IHNV could successfully infiltrate the trout’s OM, indicating that the OM could be an important portal for the IHNV. Furthermore, qPCR and RNA-Seq analysis results showed that a large number of immune-related genes were significantly upregulated in the OM of trout with IHNV infection. Critically, the results of our RNA-Seq analysis demonstrated that viral infection triggered a robust immune response, as evidenced by the substantial induction of antiviral, innate, and adaptive immune-related genes in the OM of infected fish, which underscored the essential role of the OM in viral infection. Overall, our findings revealed a previously unknown function of teleost OM in antiviral defense, and provided a theoretical basis for the study of the mucosal immunity of fish.

## 1. Introduction

The eyes play a paramount role in the lives of vertebrates, serving as vital sensory organs that enable them to interact with and perceive their surroundings [1]. They allow for the detection of light and the formation of visual images, providing crucial information for survival, navigation, and communication within their respective ecosystems [2]. The accessory structures of the eye serve a protective and motor function, including the nictitating membrane, plica semilunaris, ocular mucosa (OM), and extraocular muscles [3]. While most bony fish lack a nictitating membrane or plica semilunaris, some cartilaginous fish possess them [4]. Certain fish species, such as chub mackerel (*Scomber japonicus*) and flathead grey mullet (*Mugil cephalus*), have transparent adipose eyelids that can cover the eyeball for protection [5]. The OM covers the transparent membrane outside the eye and directly contacts the surrounding water environment, serving a protective function. However, fish do not have eyelids and tear glands like terrestrial animals, resulting in their eyes being constantly open [6]. Consequently, the teleost’s OM may face greater challenges in pathogen-infested aquatic environments.

In mammals, the OM is a major site for the invasion of pathogenic microorganisms [7,8,9]. When the ocular surface defense ability is weakened or external pathogenic factors are enhanced, conjunctival tissue inflammation characterized by vasodilation, exudation, and inflammatory cell infiltration is caused [10], for example, in conjunctivitis, a disease caused by viral (Adenoviruses), bacterial (Staphylococcus, Streptococcus, and Haemophilus influenzae), or chlamydia (*Chlamydia trachomatis*) infections [11,12,13]. Moreover, the eye might serve as an entry point for viral infections. For example, studies have shown that COVID-19 can infiltrate the eye through the human OM, inducing viral symptoms in ocular tissues [14]. With the development of the aquaculture industry, eye diseases in ornamental and intensively farmed fish are also increasing. The most common are exophthalmos (pop-eye), cataracts, keratopathy (several corneal lesions), various retinopathies, and uveitis (choroid and iris system lesions) [15,16,17]. Whether the OM is an important site of infection and invasion in other epidemic diseases has not been evaluated.

The OM of vertebrates can elicit diverse immune responses upon encountering different pathogens and antigens [18,19]. Research has shown that exposure of the OM to the infectious laryngotracheitis virus leads to altered proportions of leukocyte subsets in the head-associated lymphoid tissue and trachea of six-week-old White Leghorn chickens [20]. In a study on mucosal vaccination against avian influenza, chickens primed and boosted via the OM route using a replication-deficient recombinant adenovirus vaccine generated systemic antibodies [21]. Limited evidence also suggests there is a presence of immune responses in the OM of bony fish. For instance, a significant increase in CD45 expression was observed in the central cornea of lumpfish (*Cyclopterus lumpus*) following *Vibrio anguillarum* infection [22]. Immunological responses to vaccination have been demonstrated in various regions of the Atlantic salmon (*Salmo salar*) ocular cells expressing major histocompatibility class II [23]. These findings imply that the OM of bony fish may possess immune-protective functions.

We previously demonstrated that IHNV can infect the digestive system of juvenile rainbow trout, causing immune responses and dysbiosis in the skin, oropharyngeal mucosa, and gut [24,25,26,27]. However, IHNV infection pathogenesis in juvenile rainbow trout OM is poorly understood, as well as the innate and adaptive immune responses in OM after infection. In this study, we discovered that the virus could successfully migrate to the OM and elicit a strong immune response. Furthermore, transcriptome analysis revealed that antiviral-related genes, as well as innate and adaptive immunity-related genes, are involved in OM. These findings show the immune response and dynamics of trout OM after IHNV infection, highlighting the specificity of OM immunity following virus invasion. Collectively, our findings provide the first insight into the interaction of the OM immune system with a virus, laying the groundwork for the development of targeted immune defense strategies.

## 2. Results

### 2.1. Establishment of IHNV Virus Infection Model in Rainbow Trout

Here, we developed an infection model so that rainbow trout were infected with IHNV via immersion (Figure 1a). Infected rainbow trout exhibited characteristic clinical symptoms such as eye protrusion, pale gills, and scale loss (Figure 1b). During the experiment, approximately 37% of the infected fish died within the initial two weeks postinfection (Figure 1c). Using qPCR, we examined the dynamic change of viral load in various tissues, the results revealing a consistent trend of viral load among the OM, spleen, and head kidney (Figure 1d–f). In brief, after virus infection, virus replication peaked at 4 days postinfection (DPI) in the tissues examined, then gradually decreased until 28 DPI. Moreover, employing immunofluorescence microscopy, IHNV was found to be widely present in the OM, spleen (SP), and head kidney (HK) of infected fish (Figure 1g). Additionally, at 4 DPI, the supernatant from the tissue homogenate was collected and added to carp epithelioma cells (EPC), resulting in observable cytopathic effects (CPEs) (Figure 1h). Collectively, these outcomes substantiated the successful establishment of the trout IHNV infection model, highlighting the ability of IHNV to infiltrate OM tissues and viral loads which progressively diminish over time as survival duration increased.

### 2.2. Innate and Adaptive Immunity Genes Were Induced in OM, SP, and HK after Viral Infection

Next, a reverse transcriptase quantitative polymerase chain reaction (RT-qPCR) was performed to detect the expression of 19 immune-related genes in trout OM, SP, and HK at 0.5, 1, 4, 7, 14, 21, and 28 DPI. The target genes included antiviral genes IFNAR, interferon regulatory factor 7 (IRF7), MX1, LGP2, RIG-I, STAT1, MDA5, tripartite motif 25 (TRIM 25), and IRF3, inflammatory genes (IL-10α, IL-1β, IL-2, IL-6, IL-8, and TNFR2), immunoglobulin (Ig) genes (IgM, IgT, and pIgR), and antigen presentation-related gene (MHC-II). Our results indicated that the expression of all antiviral and inflammatory genes in various tissues was significantly upregulated at 4 and 7 DPI (Figure 2a–c). Notably, compared to SP and HK, the expression of antiviral-related genes in the OM was more rapid and pronounced at 4 DPI, which was likely linked to the initial viral exposure of the OM. At 28 DPI, immunoglobulin-related genes exhibited gradual upregulation, with conspicuous increases in the expression of IgM and IgT within the OM. This response was likely aimed at enhancing specific immune responses, neutralizing the virus, and promoting virus clearance. These findings underscore the significant role played by OM in the antiviral process. Furthermore, the relative expressions of IgT, IgM, and pIgR within the OM, SP, and HK were analyzed over time at 0 (control), 4, and 28 DPI (Figure 2d–f), revealing a sharp surge in antibody expression levels at 28 DPI, particularly within the OM. Interestingly, pIgR expression had already intensified as early as 4 DPI. These results suggest that IHNV infection could activate immune responses in both mucosal and systemic tissues, and highlight the potential role of OM in defending against viral invasion. Moreover, in terms of immune response correlations, 4 DPI and 28 DPI displayed the strongest relevance (Figure 2a–c). Therefore, these two time-points were selected for RNA-Seq analysis to further comprehensively understand the immune response unfolding in OM tissues following IHNV infection.

### 2.3. Dentification of the Transcriptome in the OM after IHNV Infection

After sequencing, we obtained a comprehensive transcriptomic profile from the trout OM. RNA-seq analyses were conducted on the Illumina platform using samples from two time-points (4 and 28 DPI). These profiles were cross referenced with the trout reference genome, using threshold filters to eliminate duplicates. Prior to the identification of differentially expressed genes (DEGs), a sample correlation analysis was executed on the filtered genes to ensure consistency among biological replicates and to visually map out the correlations across various OM groups (Figure 3a). This analysis revealed strong correlations within replicate groups, distinguishing them from other groups. Next, the expression patterns of DEGs within the Con/4 DPI and Con/28 DPI OM groups were analyzed via volcano plots, respectively (Figure 3b). Using a false discovery rate (FDR) threshold of <0.05 and |log2 Fc| ≥ 1, a total of 2993 genes in the 4 DPI group and 8431 genes in the 28 DPI group were found to be differentially regulated in the OM in response to IHNV infection. Specifically, at 4 and 28 DPI, 2709 genes and 3733 genes were upregulated, whereas 584 genes and 3698 genes were downregulated, respectively. Moreover, 1301 genes were jointly regulated at 4 and 28 DPI, including 1099 shared upregulated genes and 78 shared downregulated genes (Figure 3c). These unique and shared DEGs were focused on in the following analysis.

### 2.4. Functional Annotation and Classification of Genes

Using the Blast2GO suite, the result of gene ontology (GO) analysis showed that the DEGs within the trout OM at 4 and 28 DPI were concentrated on the following items: biological processes, cellular components, and molecular functions (Figure 4a,b). Within the biological process category, “cellular process” and “single-organism process” were the dominant groups. Within the cellular component category, “membrane” and “cell” were the most abundant groups. Within the molecular function category, “binding” and “catalytic activity” were the dominant groups. Moreover, gene function annotation based on the Kyoto Encyclopedia of Genes and Genomes (KEGG) database was used to map the DEGs within the trout OM at 4 and 28 DPI to six specific pathways, including cellular processes, environmental-information processing, genetic-information processing, human disease, metabolism, and organismal systems (Figure 5a,b). These annotated genes were further grouped into 50 two-level subclass pathways. At 4 DPI, the most substantial subcategory, “cell adhesion molecules,” encompassed 192 annotated genes, followed by “MAPK signaling pathway” (188), “herpes simplex virus 1 infection” (176), “cytokine-cytokine receptor interaction” (150), “NOD-like receptor signaling pathway” (146), and “salmonella infection” (144) (Figure 5a). On the other hand, at 28 DPI, the most extensive subcategory, “herpes simplex virus 1 infection,” comprised 176 annotated genes, followed by “NOD-like receptor signaling pathway” (125), “cytokine-cytokine receptor interaction” (111), “necroptosis” (81), “cell adhesion molecules” (76), and “salmonella infection” (60) (Figure 5b). These uniquely expressed DEGs in different groups might represent the onset/termination of specific physiological processes following IHNV infestation.

### 2.5. Classification and Analysis of DEGs

In addition to the gene functions discussed above, we focused on biological processes related to signal transduction, host defense response, and immune response in GO enrichment analysis. As illustrated in the bubble plots depicted in Figure 6, various innate immunity-related pathways were significantly enriched in both the 4 and 28 DPI groups, with type I interferon responses mainly enriched at 4 DPI, whereas apoptosis and immune cell activation were mainly enriched at 28 DPI. These results indicated that the OM mounts a robust antiviral response in the early stages of IHNV infection, simultaneously stimulating apoptosis in compromised cells and eliciting an adaptive immune cell response toward the end of the infection period. A total of 30 representative immune-related genes were then selected from the detected DEGs, and the gene expression levels of the OM at 4 and 28 DPI after IHNV infection were analyzed. The target genes included those associated with antiviral responses (MX3, MX1, IRF7, IRF3, DHX58, IFIT5, IFN-β, IFN-γ2, TRIM25, MDA5), innate immunity (CCL19; CALCOCO2; GBP1; C3; TLR3; CCL11; PXDN; MyD88; TNFRSF1A; IL-1β), and adaptive immunity (IgM, CD22, CD83, CD276, MHC-I, CD3, CD86, CD4, SASH3, pIgR) (Figure 6c–e). Among these DEGs, genes involved in antiviral responses and innate immunity were significantly upregulated at 4 DPI, whereas genes associated with adaptive immunity were significantly upregulated at 28 DPI. These observations highlight the robust antiviral response of OM tissues via innate immunity during the initial stages of disease onset. As the viral load approaches its lowest level by 28 DPI, adaptive immunity-related genes began to play a more prominent role in preparation for future pathogen invasion.

### 2.6. Enrichment and Analysis of Pathways after Infection

Then, immune genes screened out from 1 and 14 DPI groups were analyzed using the KEGG databases. The category containing the most significant pathways at 4 DPI (Figure 7a) and 28 DPI (Figure 7b) was the “NOD-like receptor signaling pathway”. At 4 DPI, the most critical pathways included viral triggering and antigen presentation. In contrast, the most pivotal pathway at 28 DPI was the T/B cell receptor signaling pathway. Next, a heatmap of all DEGs was generated to visualize the expression patterns of DEGs within the “Apoptosis,” “Necroptosis,” and “Ferroptosis” pathways (Figure 7c). Moreover, an additional heatmap was generated to focus on the top 8 DEGs in these pathways. In the “Apoptosis” pathway, CASP7 and FADD-like apoptosis regulator (CFLAR) were significantly downregulated, whereas CASP3/8 exhibited upregulation at 28 DPI. Within the “Necroptosis” pathway, activation of the JAK/STAT pathway was evident, alongside decreased expression of TNFRSF6 and RBCK1. Within the “Ferroptosis” pathway, FTM was downregulated, whereas HO and CYBB were upregulated (Figure 7d). Combined, transcriptome results indicate that viral invasion disrupts the immune homeostasis of OM. Over time, by combining innate and acquired immune responses, pathogens and apoptotic cells are eliminated, and a new balance was established.

### 2.7. Enrichment and Analysis of Pathways after Infection

To study the reliability of the transcriptome data results, qPCR was conducted to verify the results. We randomly selected five upregulated genes and five downregulated genes from the transcriptome results of the 4 and 28 DPI groups, and detected their expression levels with specific primers. The results showed that the results of qPCR and transcriptome were in good agreement and indicated that the results of transcriptome analysis have high reliability (Figure 8a,b).

## 3. Discussion

Lacking lacrimal glands and eyelids, teleost fish keep their eyes constantly open [28]. In the pathogen-infested aquatic environment, the OM serves as the first line of defense to protect the eyes, which face greater pressure [29]. In mammals, ocular mucosal immunity plays a crucial role in maintaining ocular function [18,19,30]. When the function of ocular mucus weakens and environmental pressure becomes excessive, the OM can also become an important entry point for pathogenic microorganisms [31]. However, the risk of viral transmission through the OM in teleost fish has not been thoroughly evaluated, and the immune response in the OM remains unclear. Therefore, we used transcriptomics to study the dynamic changes of the OM of rainbow trout after IHNV infection.

Here, we established an infection model of IHNV through immersion. Infected trout exhibited bulging eyes and small areas of bleeding around the eyes, as well as other noticeable clinical symptoms such as pale gills and darkened skin [32]. Using qPCR and immunofluorescence, we discovered that the IHNV-*N* gene had a high level of expression in the OM, and IHNV could be detected within the OM. Previous studies have shown successful invasion of IHNV in the gut, pyloric ceca, kidney, brain, muscle, heart, skin, spleen, and liver [33]. Our results indicate that IHNV is capable of invading the OM, providing new insights into how IHNV spreads and invades. Additionally, the intrusion of the virus triggers significant immune responses within the OM, leading to an increased expression of various genes associated with immune regulation, including genes related to antiviral responses, inflammation, and immunoglobulin production. In the early phase after infection (4 days postinfection or 4 DPI), the OM quickly activates multiple antiviral and inflammatory genes to counter the viral invasion. Importantly, compared to the spleen and head kidney, the rainbow trout OM shows a more rapid and robust response at 4 DPI, potentially due to its early encounter with the virus during the initial stages of infection. This highlights the important role played by the rainbow trout OM, which is a mucosal component, in the survival of aquatic life. In the later stage after infection (28 DPI), the immune response within the OM displays different characteristics. The expression of antiviral and inflammatory genes gradually decreases, while the expression of immunoglobulin-related genes gradually increases. The main immunoglobulins involved in the mucosal immune response in fish are IgM, IgD, and IgT, with IgM being the main contributor to the systemic immune response and also present in mucosal secretions [34]. Another significant immunoglobulin, similar to mammalian IgA, is IgT, which is the only teleost Ig isotype with specialized mucosal functions [35,36]. At 28 DPI, the expression of IgM and IgT significantly increases. These antibodies enhance specific immune responses, neutralize the virus, and aid in clearing the infection. This suggests that in the later stages after infection, the OM initiates the development of adaptive immunity to better respond to future pathogen invasions. These results demonstrate that IHNV can activate the immune response of the OM, involving both innate and adaptive immunity. In the subsequent segment, we conducted further transcriptome analysis to comprehensively understand the virus-induced processes at these two specific time points.

We conducted GO and KEGG enrichment analysis on upregulated DEGs and identified a broad range of response processes in the OM triggered by the virus. These processes include cellular processes, environmental-information processing, genetic-information processing, human diseases, metabolism, and organismal systems. Notably, a significant number of genes are associated with antiviral processes. To gain further insight into the active response processes in the OM, we conducted additional analyses on the upregulated DEGs related to signal transduction, host defense responses, and immune responses. The results of the GO enrichment analysis revealed significant enrichment of various immunity-related pathways in the biological process category in the 4 and 28 DPI groups. Among these pathways, type I interferon responses were particularly enriched at 4 DPI. These responses, known for their induction of interferon-stimulated genes (ISGs), play a crucial role in impeding virus replication, degrading viral RNA, inhibiting viral RNA translation and modification, and preparing surrounding infected cells for antiviral responses [37]. Additionally, the interplay of type I interferon responses and innate immune cells helps inhibit virus replication [38]. Furthermore, genes involved in apoptosis and immune cell activation were primarily enriched at 28 DPI. These findings emphasize the robust antiviral response of the OM during the early stages of IHNV infection, followed by the induction of apoptosis in compromised cells and an adaptive immune cell response in the later stages of infection.

The results of our KEGG pathway enrichment analysis emphasized the importance of the NOD-like receptor (NLRs) signaling pathway in the OM following infection. NLRs play a crucial role in recognizing specific pathogen molecules or host-derived damage signals in the cytoplasm, leading to the activation of the innate immune response [39]. Activated NODs can also interact with autophagy-related ATG16L1 to induce autophagy [40]. Additionally, other pattern-recognition receptors, including NLRs, can be activated by different pathogens or endogenous danger signals, forming inflammasome complexes. This leads to the cleavage and activation of pro-Caspase-1, promoting the processing and secretion of precursor forms of IL-1β and IL-18, ultimately inducing apoptosis, a form of inflammatory cell death [41]. During the early stages of infection (4 DPI), the virus triggers the activation of the Toll-like receptor (TLR) signaling pathway. The ability of cells to recognize pathogen-associated molecular patterns (PAMPs) depends on the expression of the toll-like receptor family [42]. TLRs play a crucial role in initiating appropriate innate responses to viral PAMPs found in extracellular compartments or endosomes. Furthermore, the RIG-I-like receptor signaling pathway is also activated, with RIG-I and MDA5 acting as cytoplasmic sensors for double-stranded RNA (dsRNA). RIG-I specifically induces type I interferon (IFN) responses to different lengths of in vitro transcribed dsRNA, which helps to quickly fight against viral invasions [43,44]. On the other hand, during later stages of infection (28 DPI), the NF-κB pathway becomes activated. This pathway, in particular, plays a role in the differentiation, development, and survival of B cells [45]. Along with the activation of the B cell receptor pathway, these responses suggest that adaptive immunity, mainly mediated by B cells, is established at this stage. Additionally, certain cytokine pathways are involved in regulating cellular processes like apoptosis, which helps maintain tissue homeostasis and eliminate damaged or abnormal cells. Further analysis of the Apoptosis, Necroptosis, and Ferroptosis pathways at 28 DPI has provided interesting insights. The anti-apoptotic protein CFLARL inhibits apoptosis and necroptosis of T lymphocytes by blocking CASP8 activation [46]. Caspase-8 can also activate NF-κB in its unprocessed and inactive precursor form [45,47]. Based on our findings, we observed significant downregulation of the apoptosis pathway CFLAR, as well as upregulation of CASP3/8, indicating that T lymphocytes enter the apoptosis pathway in the late stage of infection, while B cells may contribute to adaptive immunity. Activation of the JAK pathway leads to the phosphorylation of other targets, including receptors and major substrates like STATs, and subsequently activates potential transcription factors in the cytoplasm. Thus, the JAK/STAT cascade provides a direct mechanism for translating extracellular signals into transcriptional responses [48]. Our results demonstrate that in the necroptosis pathway, activation of the JAK/STAT pathway coincided with decreased expression of TNFRSF6 and RBCK1. This provides a relevant mechanistic basis for the orchestrated regulation of inflammatory cell death. Furthermore, the ferroptosis pathway showed downregulation of FTM and activation of CYBB, a crucial factor for driving ferroptosis [49]. The simultaneous activation and inhibition of these pathways collectively establish a crucial foundation for cellular-level immune response and regulation in OM.

## 4. Materials and Methods

### 4.1. Fish Maintenance

All experimental rainbow trout (approximately 5 g) were purchased from an aquaculture farm in Hubei (Yichang, China), and then maintained and acclimatized in a water-recirculation system at 16 °C for at least 2 weeks. The water in the water-recirculation system was constantly aerated and filtered. Furthermore, the water condition was checked every day, including pH (6.5–8.5), nitrite/nitrate/ammonia levels (<0.1 mg/L), and oxygen levels (5–12 mg/L). During the experiment, fish were fed with commercial feed pellets twice per day.

### 4.2. Infection of Fish with IHNV and Sample Collection

The IHNV (JL14−2814) was provided by Hong Liu of Shenzhen Academy of Inspection and Quarantine Sciences. The strategy for IHNV infection in rainbow trout is the bath infection described previously. Briefly, 150 fish (approximately 5 g each) were soaked in 10 L aerated water containing 6 mL IHNV (1 × 10^9^ pfu/mL) for 2 h at 16 °C and then transferred to an aquarium filled with fresh water [24]. The number of dead fish was recorded for the statistics of cumulative survival during the experiment. Tissue samples were collected from six individuals at 0.5, 1, 2, 4, 7, 14, 21, and 28 DPI. Fish were euthanized with an overdose of tricaine mesylate MS-222 (Sigma, St. Louis, MO, USA), and tissue samples including OM, SP, and HK were collected for transcriptome sequencing, immune gene expression, and viral load. As a control group (mock infection), the same number of fish were reared in similar tanks and exposed to the same pathogen-free medium. Two experiments were performed independently at least three times.

### 4.3. Standard Curve for IHNV

The PCR product of IHNV was connected to the pMD19-T vector, and then transformed into *Escherichia coli* DH5α. The bacterial solution was cultured in a plate containing a solid medium containing ampicillin for overnight. After the single bacteria were selected for PCR identification, the positive clones were further expanded for culture. Plasmid DNA was extracted using the Plasmid Mini Kit (TIANGEN). The recombinant plasmid was serially diluted 10 times (a total of 7 gradients from 1.3 × 10^8^ copy/mL to 1.3 × 10^2^ copy/mL). Then, a standard curve of IHNV was made using the Ct values and copy numbers.

### 4.4. RNA Isolation and qPCR Analysis

Various tissues were sampled into 2 mL EP tubes with 1 mL trizol (Invitrogen, Carlsbad, CA, USA), and RNA was extracted following the manufacturer’s protocol. The concentration of every samples’ RNA was determined using spectrophotometry Nanodrop ND1000 (LabTech, Beijing, China), and the integrity of RNA was examined using 1% agarose gel electrophoresis. For each sample, the equal mass of total RNA (1000 ng) was used for cDNA synthesis using the Hifair III 1st Strand cDNA Synthesis SuperMix for the qPCR set (YEASEN, Shanghai, China). Next, the cDNA was diluted three times for further qPCR analysis. qPCR was performed using the qTOWER3G PCR system (Analytik Jena, Jena, Germany) using Hieff qPCR SYBR GREEN Master Mix (YEASEN) under the following conditions: 95 °C for 5 min, then cycled 40 times at 95 °C for 10 s, at 58 °C for 30 s [50]. The qPCR primer sequences used in this experiment are provided in Table 1.

### 4.5. Immunofluorescence Studies

For IHNV detection, the OM, SP, and HK tissues of rainbow trout were fixed in 4% neutral buffered formalin, and then dehydrated and paraffin embedded. Furthermore, 5-μm tissue sections were used for immunofluorescence. Briefly, after blocking with StartingBlock™ Blocking Buffer (Thermo Fisher Scientific, Waltham, MA, USA) at room temperature for 30 min, sections were incubated with mouse anti−IHNV−N mAb (1 μg/mL; BIO−X Diagnostics, Rochefort, Belgium) overnight at 4 °C. After washing four times with PBS, the sections were stained with Cy3-conjugated AffiniPure goat an-ti-mouse IgG pAb (3 μg/mL) for 45 min at room temperature. Then, DAPI (4′, 6-diamidino-2 phenylindole; 1 μg/mL; Invitrogen, Carlsbad, CA, USA) was used to stain the nucleus of tissue cells. All images were acquired using an Olympus BX53 fluorescence microscope (Olympus, Shinjuku City, Tokyo, Japan).

### 4.6. Plaque Assay of IHNV

EPC were cultured in Medium 199 (Gibco, New York, NY, USA) supplemented with 10% fetal bovine serum (FBS) (Gibco). The EPC was maintained at 28 °C in an atmosphere containing 5% CO_2_. IHNV was isolated from supernatants obtained after grinding individual tissues of infected fish. The cell suspension was centrifuged for 5 min at 400× *g* once and 10,000× *g* thrice at 4 °C to remove cell debris, and then, the supernatant was filtered with a 0.22 μm membrane filter to remove bacteria in the mucosal tissue. When CPEs were clearly apparent, images of infected cells were captured using a microscope.

### 4.7. RNA-Seq Library Construction, Sequencing, and Data Analyses

Ocular mucosal samples in 4 DPI/control and 28 DPI/control were submitted to Majorbio Technology Co., Ltd. (Wuhan, China) for analysis. The overall process mainly includes the following steps: RNA was extracted from tissue samples using a trizol kit, and Illumina TruseqTM RNA sample-prep-kit methods were used for library construction after quality inspection. The mRNA was enriched using oligo dT, followed by fragmentation, cDNA synthesis, adaptor ligation, and sequencing on a HiSeq X Ten sequencer (Illumina^®^, San Diego, CA, USA). The reads obtained from the sequencer were mapped to the rainbow trout genome using STAR (version 2.5.3a) and were counted by feature using Subread-1.5.1 in the Bioconductor package [50,51,52,53]. Furthermore, differentially expressed genes (DEGs) were identified using the edgeR package (version 3.12.1). GO and KEGG enrichment analyses were performed using KOBAS (version: 2.1.1) [50,54].

### 4.8. Statistical Analysis

All data shown in this paper were presented as mean ± standard deviation (SD). SPSS software (version 18.0) was used with statistical analysis, and an unpaired Student’s *t*-test was used to calculate the significant difference between the data (*p* < 0.05).

## 5. Conclusions

In sum, our findings represent a novel investigation into the antiviral immune response of trout’s OM. In this study, IHNV was capable of invading the trout OM. Furthermore, IHNV infection elicited a robust antivirus and immunoregulation response in the trout OM, characterized with the upregulation of antiviral genes, inflammatory genes, adaptive immunity-related genes, notably chemokine, inflammatory cytokines, and immunoglobulin. Significantly, these findings highlight the importance of OM on mucosal immune defense function in trout and its potential contribution to mucosal immunity research. Nonetheless, the immune function of the ocular mucosa of bony fish remains to be further studied, especially the complex immunomodulatory process in the process of resisting foreign pathogens.

## Figures and Tables

**Figure 1 ijms-25-00160-f001:**
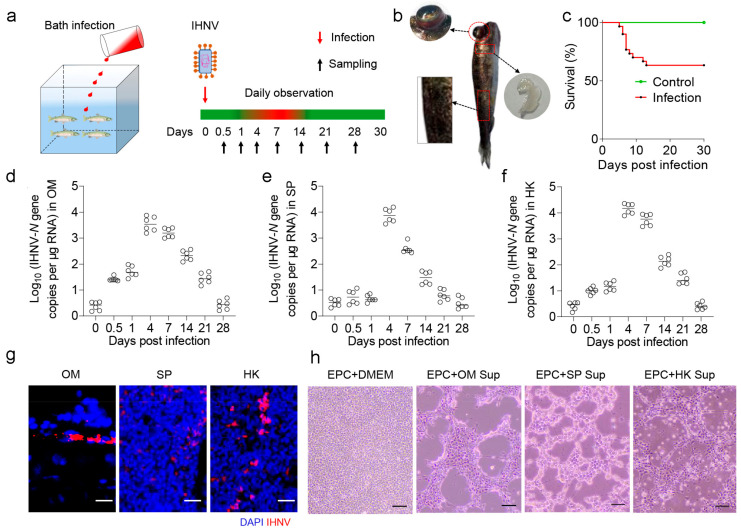
Trout infection model of IHNV. (**a**) Trout were infected through IHNV baths for 4 h, and then sacrificed at 0.5, 1, 4, 7, 14, 21, and 28 DPI to collect tissue samples. Red on the timeline represents dead or diseased fish with symptoms of viral infection (i.e., slower swimming and less-active feeding compared to control fish), while green represents the normal behavior of healthy fish. (**b**) Basic symptoms of diseased fish infected with IHNV. (**c**) Cumulative survival of control and IHNV-infected fish. IHNV-N gene copies were quantified using qPCR in fish tissues collected at 0.5, 1, 4, 7, 14, 21, and 28 DPI. The virus was detected in the fish OM (**d**), spleen (SP) (**e**), and head kidney (HK) (**f**). (**g**) Immunofluorescence staining of IHNV in OM, SP, and HK from 4 DPI infected fish (*n* = 6). IHNV (red) is stained with an anti-IHNV-*N* mAb; nuclei were stained with DAPI (blue). Scale bars, 20 μm. (**h**) Cytopathic effect of IHNV on EPC cells after incubation with OM, SP, and HK homogenate supernatants from infected fish at 4 DPI. Scale bars, 100 μm.

**Figure 2 ijms-25-00160-f002:**
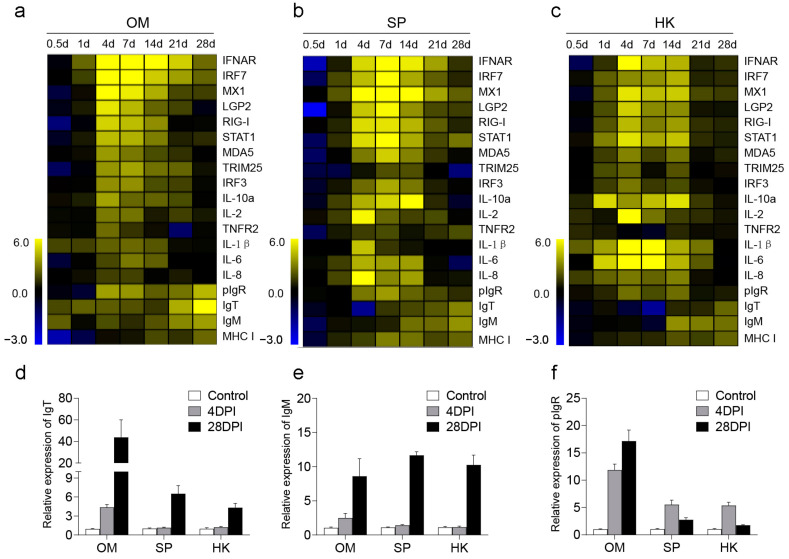
Expression of immune-related genes in trout OM, SP, and HK after IHNV infection. Heatmaps illustrate results from qPCR of transcripts for selected immune markers from several lymphoid organs of IHNV-infected versus control fish measured at 0.5, 1, 4, 7, 14, 21, and 28 DPI, OM (**a**), SP (**b**), HK (**c**) (*n* = 6). Relative expression of IgT, IgM, and pIgR in each tissue at control, 4 and 28 DPI, IgT (**d**), IgM (**e**), and pIgR (**f**).

**Figure 3 ijms-25-00160-f003:**
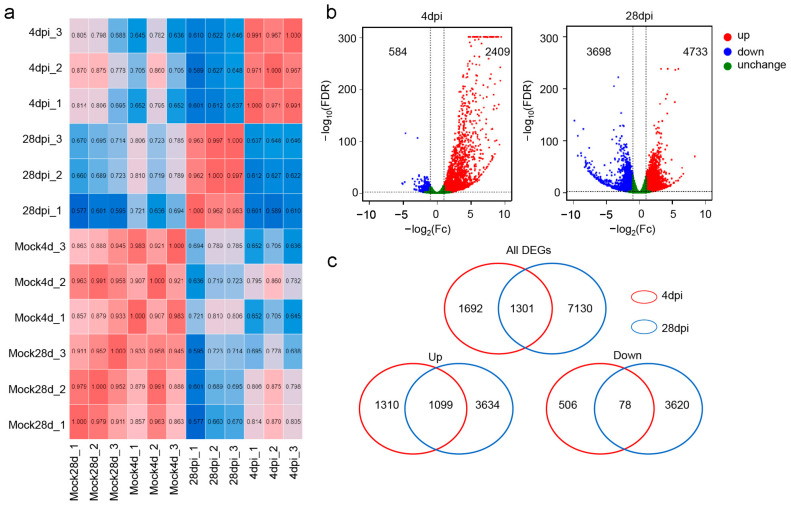
Cluster and DEGs analysis of trout OM transcriptome after infection with IHNV (*n* = 3). (**a**) Sample correlation analysis graph in different groups. The horizontal and vertical coordinates are all samples, and the samples that are clustered together are similar samples. The number represents the correlation coefficient, and the color from blue to red indicates that the correlation is from low to high, and the redder it is, the more relevant it is. (**b**) The volcano plots of the DEGs distribution. The DEGs of OM in trout at 4 DPI (left) and 28 DPI (right) compared to controls. Red spots, upregulated expression DEGs. Blue spots, downregulated expression DEGs. Green spots indicate no difference. (**c**) Venn diagram of the number of DEGs. The red circle represents the group after 4 DPI, and the blue circle represents the group at 28 DPI.

**Figure 4 ijms-25-00160-f004:**
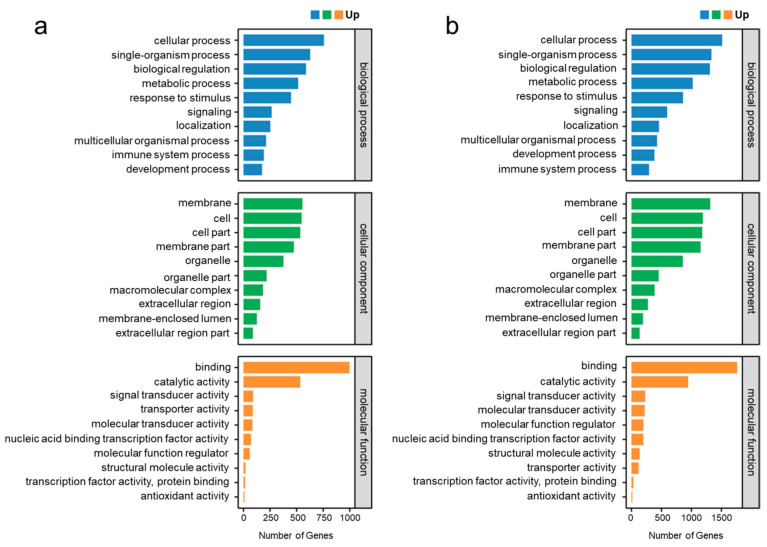
Functional enrichment analysis of GO terms for upregulation of DEGs at 4 DPI (**a**) and 28 DPI (**b**). The *x*-axis represents the biological process, and the *y*-axis represents the number of genes in the process.

**Figure 5 ijms-25-00160-f005:**
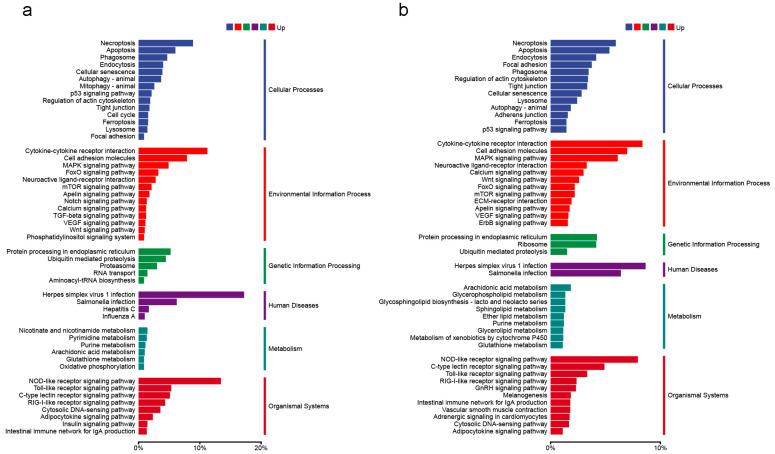
The KEGG classification of upregulation of DEGs at 4 DPI (**a**) and 28 DPI (**b**). Genes are assigned to six special KEGG pathways, including organismal systems, metabolism, human disease, genetic-information processing, environmental-information process, and cellular processes.

**Figure 6 ijms-25-00160-f006:**
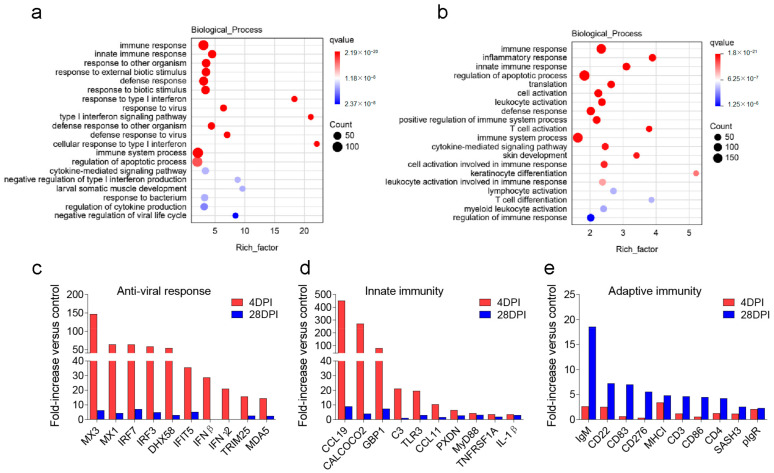
Biological process enrichment and differential immune gene analysis. GO enrichment analysis of DEGs related to signal transduction, host defense responses, and immune responses at 4 DPI (**a**) and 28 DPI (**b**). The size of the bubbles represents the number of genes in a pathway, while the color represents qvalue. The rich factor of the *x*-axis refers to (the number of DEGs annotated to the GO term)/(the total number of genes annotated to the GO term). Representative antiviral response (**c**), innate (**d**), and adaptive (**e**) immune genes modulated by IHNV infection at 4 or 28 DPI (*n* = 3). Data are expressed as mean fold increase in expression.

**Figure 7 ijms-25-00160-f007:**
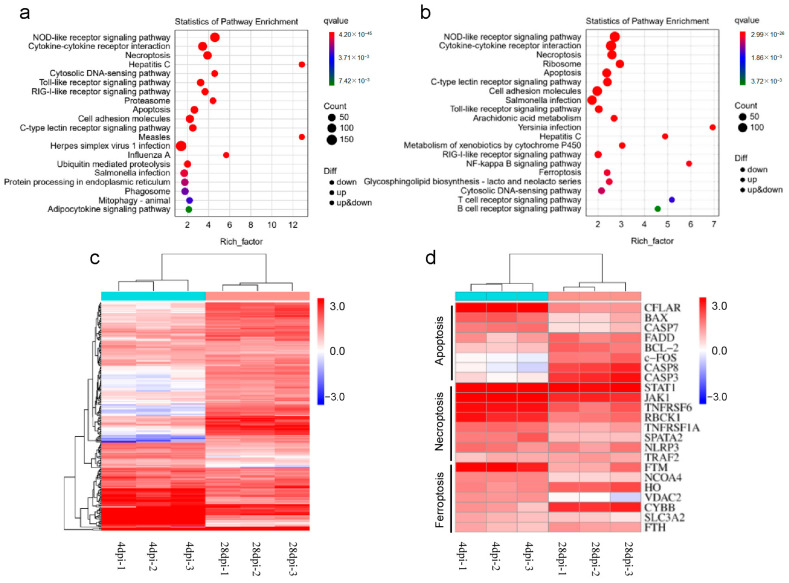
Pathway enrichment and differential gene analysis. Top 20 pathway enrichment in 4 DPI (**a**) and 28 DPI (**b**). The *x*-axis represents the number of genes in the DEGs pathway and its percentage among all genes. The *y*-axis represents the path. (**c**) A heat map of all DEGs was made for the enriched “Apoptosis”, “Necroptosis”, and “Ferroptosis” pathways. (**d**) Make a heat map with the top 8 differentially expressed genes in the above pathways.

**Figure 8 ijms-25-00160-f008:**
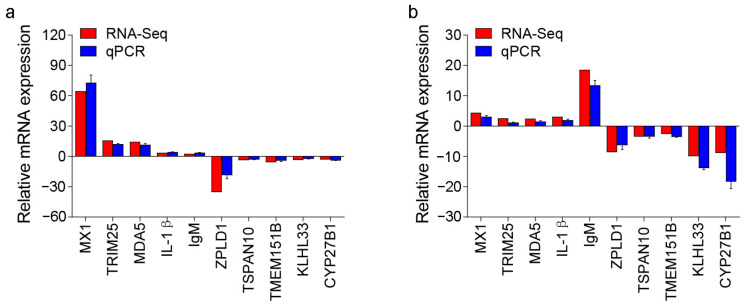
Comparison of the expression profiles of ten selected genes as determined using RNA sequencing and qPCR in 4 (**a**) and 28 DPI (**b**).

**Table 1 ijms-25-00160-t001:** Primers used in this study.

Gene	Forward Primer (5′-3′)	Reverse Primer (5′-3′)
EF1α	CAACGATATCCGTCGTGGCA	ACAGCGAAACGACCAAGAGG
MDA5	CAGTGGAGATGACGATGGG	ACTTGGCGTTCTTGTGCTT
LGP2	AGTTTGGCACGCAGGAGTA	CAAGCAGGAAGAAGTCGGT
IRF7	GATGCCTCAAAATGAAATGG	TCAGGGAACTTCTCACCAAA
IFNAR	CAGAGCCTCAGGAAGAACT	CAAGGGGTAGAAGAGCATA
RIG-I	CAGAGGTACTACAGGAAATGG	TTACTGGTCTTCAAGCAATG
STAT1	CTCATCCCCTGGACCAAGTT	TTATTGTAGCCCTCCACCCA
TRIM25	AAAGATTCACCCCAAAACC	AAGGCAGGGGAATCATAGT
IRF3	GGAGACCACTCTGTTTGGAAG	CGGCATCGTTCTTGTTGTC
IL-10a	CGAAGGGATTCTACACCACTTG	GAGATTTAAAGTCGTTGTTGTTCTG
IL-2	CGAACGGGACGAGAAATGG	TGATAAAGAGCTGCTGTGAATG
TNFR2	GGAAAGGAGGCCATTAAATGTCTGC	CAGATGTCAGAGGGCTGGTGGA
pIgR	GATGACCCAGAATCACCCCC	GTCAGACACCCACATTCCCC
ZPLD1	AAAGCCACAAACCTG	TTGACTCTTCCCATTC
TMEM151B	TGCGGTCAAATCGTC	ATGGCAGAATCAAGTCA
KLHL33	AGGTATTCCAGGGTCT	GAACAAGGGTACTCACA
TSPAN10	AACCTCCTCTTCACGG	CCACAAACACCAGCAT
CYP28B1	AGGCTGTGGTCAAGG	GTGGCAGAGGGTGAT
IgM	AAGAAAGCCTACAAGAGGGAGA	CGTCAACAAGCCAAGCCACTA
IgT	CAGACAACAGCACCTCACCTA	GAGTCAATAAGAAGACACAACGA
IL-6	GCTCGGGACAGACGGTGAAA	GAAACTCCTCCACAAACTGTTG
IL-8	GGCCCTCCTGACCATTACTGA	TCCAGACAAATCTCCTGACCG
IL-1β	ACCGAGTTCAAGGACAAGGA	CATTCATCAGGACCCAGCAC
MHC-I	AGGCTGGAGTCACAGAGGG	CGTGGCAGGTCACTGGAGA
MX1	GATGCTGCACCTCAAGTCCTACTA	CGGATCACCATGGGAATCTGA

## Data Availability

The raw RNA sequencing data have been deposited in the NCBI Sequence Read Archive under BioProject accession number PRJNA1012685.

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
