# Peer review of "Teleost Eye Is the Portal of IHNV Entry and Contributes to a Robust Mucosal Immune Response"

_ijms, 2023, doi:10.3390/ijms25010160_

Round 1
Reviewer 1 Report
Comments and Suggestions for Authors
This research offers significant contributions to understanding the function of teleost OM in antiviral defense, providing valuable insights for future research into the mechanisms and functions of teleost OM immunity. The abstract and introduction well-described background by placing the question address and emphasizing the purpose of this study. The research has a high scientific level, the experiment is well-designed, the discussion is consistent with the results, and the final conclusions are interesting. Therefore, I recommend this manuscript for publication after improving some minor points.
Material and method
- Authors should add a declaration statement of ethical approval for this study.
- Does each primer set have the same annealing temperature of 58 degrees Celsius?
Line 418; How long does it take to clearly display CPEs?
Line 421; The author should provide a short explanation for the selection of these time points which correlates with the result from 2.1.
Author Response
Please see the attached file for the responses to reviewer's comments.

Reviewer 2 Report
Comments and Suggestions for Authors
The manuscript ijms-2742925 describes the immune response present in the ocular mucosa of rainbow trout after infection with bath-administered IHNV.
The paper is interesting and well constructed, although not always easy to understand. I don't see any major problems in this draft.
It remains to clearly insert the chemical-physical parameters of the water during the experimentation; furthermore, on line 393 write the indicated bacterial species in italics. Finally, on line 412 a bibliographic citation is inserted which is not included in the references (Morgan et al.) and which must also be reported correctly in the text.
For these reasons and the importance of the manuscript, in my opinion, this article can be published after minor revision.
Author Response
Please see attached file for the responses to reviewer's comments.

Reviewer 3 Report
Comments and Suggestions for Authors
Review comments: In this manuscript, the authors reported “Teleost Eye as the Portal of IHNV Infection and Mount a Robust Mucosal Immune Response”. It is meaningful to provide new insights into the pivotal role of the ocular mucosa (OM) in antiviral immunity among teleost fish, which contributes to the further research into the mechanisms and functions of OM immunity in early vertebrate species. It is considered that this research merits publication in International Journal of Molecular Science. There are few comments which may be considered for the improvement of the manuscript before publication:
1. There is no description about the statistics of cumulative survival in the section of Materials and Methods.
2. The methods of Immunofluorescence staining of IHNV are missing.
3. Minor comments:
(1) In line 94, the full name of the abbreviations (SP and HK) should be provided at first appearance.
(2) In line 393, “Escherichia coli” should be italic.
(3) It’s better to add the origin of IHNV.
Author Response

(The authors gave the same response as above.)
